# Facile Strategy for Boosting of Inorganic Fillers Retention in Paper

**DOI:** 10.3390/polym16010110

**Published:** 2023-12-29

**Authors:** Klaudia Maślana, Krzysztof Sielicki, Karolina Wenelska, Tomasz Kędzierski, Joanna Janusz, Grzegorz Mariańczyk, Aleksandra Gorgon-Kuza, Wojciech Bogdan, Beata Zielińska, Ewa Mijowska

**Affiliations:** 1Department of Nanomaterials Physicochemistry, Faculty of Chemical Technology and Engineering, West Pomeranian University of Technology in Szczecin, Piastow Ave. 45, 70-311 Szczecin, Poland; krzysztof-sielicki@zut.edu.pl (K.S.); tomasz.kedzierski@zut.edu.pl (T.K.); bzielinska@zut.edu.pl (B.Z.); emijowska@zut.edu.pl (E.M.); 2Arctic Paper Kostrzyn SA, ul. Fabryczna 1, 66-470 Kostrzyn nad Odra, Poland; joanna.janusz@arcticpaper.com (J.J.); grzegorz.marianczyk@arcticpaper.com (G.M.); wojciech.bogdan@arcticpaper.com (W.B.)

**Keywords:** cellulose, crosslinker, fillers, polymer

## Abstract

Achieving the desired properties of paper such as strength, durability, and printability remains challenging. Paper mills employ calcium carbonate (CaCO_3_) as a filler to boost paper’s brightness, opacity, and printability. However, weak interaction between cellulose fibers and CaCO_3_ particles creates different issues in the papermaking industry. Therefore, this study explores the influence of various inorganic additives as crosslinkers such as mesoporous SiO_2_ nanospheres, TiO_2_ nanoparticles, h-BN nanoflakes, and hydroxylated h-BN nanoflakes (h-BN-OH) on inorganic fillers content in the paper. They were introduced to the paper pulp in the form of a polyethylene glycol (PEG) suspension to enable bonding between the inorganic particles and the paper pulp. Our findings have been revealed based on detailed microscopic and structural analyses, e.g., transmission and scanning electron microscopy, X-ray diffraction, Raman spectroscopy, and N_2_ adsorption/desorption isotherms. Finally, the inorganic fillers (CaCO_3_ and respective inorganic additives) content was evaluated following ISO 1762:2001 guidelines. Conducted evaluations allowed us to identify the most efficient crosslinker (SiO_2_ nanoparticles) in terms of inorganic filler retention. Paper sheets modified with SiO_2_ enhance the retention of the fillers by ~12.1%. Therefore, we believe these findings offer valuable insights for enhancing the papermaking process toward boosting the quality of the resulting paper.

## 1. Introduction

The process of papermaking involves converting raw materials, like wood fibers or recycled paper, into a product that meets quality standards and market demands. One of the main challenges in this process is achieving the desired properties in the final product, such as strength, durability, brightness, opacity, and printability, which enhance the strength and stiffness of the paper, avoiding poor folding, tearing, and cracking resistance. To achieve this, paper mills often use fillers such as CaCO_3_ [1]. However, the weak interaction between cellulose fibers and CaCO_3_ particles can lead to a range of problems in papermaking [2,3]. This problem is related to the cellulose fibers’ structural properties, which have a highly crystalline structure with a low surface-area-to-volume ratio. As a consequence, a limited availability of active sites for hydrogen bonds, both between fibers and with CaCO_3_ particles, is observed [4,5]. The interaction between cellulose fibers and CaCO_3_ particles affects paper mass drainage and retention. The presence of fillers can interfere with water flow through the paper machine, leading to longer drying times and increased energy consumption [6].

Various strategies have been developed to overcome these obstacles to enhance the affinity of the paper mass to CaCO_3_. One of the strategies is the addition of crosslinkers, which can improve the interaction between the cellulose fibers and CaCO_3_ particles. Crosslinkers are chemical compounds that can form bonds between the cellulose fibers, creating a more stable network that can better resist the disruptive effects of fillers [7,8]. The addition of fillers such as SiO_2_ [9], TiO_2_ [10], h-BN [11], and h-BN-OH [12] has been shown to be effective in improving the affinity of the paper mass to CaCO_3_. These compounds can form bonds with both the cellulose fibers and CaCO_3_ particles, creating a stronger network that improves the strength, stiffness, and printability of the paper. Therefore, using crosslinkers in papermaking is an important strategy for improving the quality and performance of paper products.

Silicon dioxide (SiO_2_) can form covalent bonds with both the cellulose fibers and CaCO_3_ particles. The SiO_2_ particles can also serve as a connector between the cellulose fibers and CaCO_3_ particles, helping to strengthen the inter-fiber bonding and improve the stiffness and strength of the paper. In addition to its role as a crosslinker, SiO_2_ can also improve the drainage and retention of the paper mass. The SiO_2_ particles are hydrophilic, which means that they can help absorb water and improve the flow of the paper mass through the paper machine. This can lead to faster drying times and increased productivity. SiO_2_ can be added to the paper mass in various forms, including colloidal silica, precipitated silica, and silica fume [9,13]. 

Titanium dioxide (TiO_2_) as a crosslinker can also improve the optical properties of paper products. TiO_2_ is a white pigment that can reflect and scatter light, leading to a brighter and more opaque paper. This can be particularly beneficial in applications where high brightness and opacity are expected. In addition, TiO_2_ can also improve the drainage and retention of the paper mass—similar to SiO_2_. This can lead to faster drying times and increased productivity. TiO_2_ can be added to the paper mass in various forms, including rutile, anatase, and nano-sized TiO_2_ particles. Rutile and anatase are two crystalline forms of TiO_2_, with rutile being the most commonly used form in papermaking due to its higher refractive index and opacity. Nano-sized TiO_2_ particles have a smaller particle size and can provide additional benefits such as improved printability and ink adhesion [10,14].

Hexagonal boron nitride (h-BN) is a layered material that consists of hexagonally arranged boron and nitrogen atoms. It has a high surface area and unique surface chemistry that make it attractive for various applications. When added to the paper mass, h-BN can form covalent bonds with both the cellulose fibers and CaCO_3_ particles, improving the interaction between them and creating a more stable network. The h-BN particles can also serve as a connector between the fibers and fillers, enhancing the stiffness and strength of the paper. h-BN-OH is a derivative of h-BN that has been functionalized with hydroxyl groups. The hydroxyl groups increase the surface energy and wettability of the h-BN particles, allowing them to interact with the cellulose fibers and CaCO_3_ particles more effectively. The hydroxyl groups can also provide additional chemical functionality, allowing for further modifications and improvements in the paper properties [12,15].

In this work, the impact of different inorganic fillers on calcium carbonate content in the paper was investigated. For this reason, mesoporous SiO_2_ nanoparticles, TiO_2_ nanoparticles, h-BN nanoflakes, and hydroxylated h-BN nanoflakes (h-BN-OH) have been explored as additives. They have been introduced to the paper pulp in the form of a polyethylene glycol (PEG) mixture to induce bonding between the inorganic structures and paper pulp components. Various characterization methods were employed to determine the chemical structure and morphology of prepared samples, including TEM, SEM, XRD, Raman spectroscopy, and TGA. The ash content (residual solid particles after the combustion process) was evaluated according to ISO 1762:2001. It allowed us to select mesoporous silica as the most efficient filler, enhancing the retention of the fillers by 12.1% in respect to unmodified paper sheets. 

## 2. Experimental Part

### 2.1. Chemicals

Eucalyptus-derived cellulose fibers, wood-derived cellulose fibers, chemical thermomechanical pulp cellulose (CTMP), cationic starch, calcium carbonate, and potassium amyl xanthate (PAX) were derived by Arctic Paper Kostrzyn SA. Polyethylene glycol 2000 (PEG, M = 1800–2200 g/mol) was purchased from Carl Roth, Karlsruhe, Germany. Titanium dioxide was kindly supplied by Grupa Azoty Police (Police, Poland). Tetraethyl orthosilicate (TEOS) and boron nitride (BN, ~1 µm, 98%) were purchased from Sigma Aldrich (Poznań, Poland). Ammonia solution (NH_4_·OH, 25%), sulfuric acid (H_2_SO_4_, 95%), nitric acid (HNO_3_, 65%), and potassium permanganate (KMnO_4_) were delivered from Chempur (Piekary Śląskie, Poland). 

### 2.2. Synthesis of Silicon Dioxide (SiO_2_)

SiO_2_ nanoparticles were prepared according to a modified Stöber method [16]; 5.8 mL of TEOS and 150 mL of ethanol were initially mixed in a round bottom flask and stirred for 10 min at room temperature (RT). Next, 7.8 mL of ammonia solution was added, and the mixture was further stirred for 12 h. After that, the product was separated by centrifugation, washed a few times with ethanol, and dried. 

### 2.3. Synthesis of Hexagonal Boron Nitride (h-BN)

A total of 0.2 g of bulk boron nitride was placed in a flask, and then, 200 mL of ethanol was added. Afterward, the obtained mixture was sonicated using an ultrasonic homogenizer for 12 h. The final product was washed with distilled water and dried at 80 °C.

### 2.4. Synthesis of Oxidized Hexagonal Boron Nitride (h-BN-OH)

To prepare h-BN-OH, a modified Hummers method was applied [12]. Briefly, 0.2 g of h-BN powder was placed in a round-bottomed flask, and then, 13.5 mL of H_2_SO_4_ and 4.4 mL of HNO_3_ were added. The mixture was stirred to obtain a homogenous dispersion. After that, 1.2 g of KMnO_4_ was partially introduced, and finally, the mixture was heated to 90 °C and kept at this temperature for 12 h. Next, the mixture was cooled to RT, filtrated, and washed a few times with distilled water until the pH value approached 7. Finally, the product was dried at 80 °C overnight. 

### 2.5. Preparation of PEG Solution

In the experiments, the different polyethylene glycol (PEG) variants (PEG 2000, PEG 4000, PEG 10000, and PEG 20000) were tested to choose the optimal one. During the preparation of paper sheets using the Rapid-Köthen machine (Lodz, Poland) different performance characteristics of the paper with the tested PEG variants were evaluated. The final selection of PEG 2000 was based on its superior performance during the paper sheet preparation process. PEG 2000 exhibited the best solubility in the chosen solvent and its integration with paper pulp. An appropriate PEG 2000 amount was added to 1 L of distilled water, and with the use of a magnetic stirrer, it was completely dissolved. The amount of PEG 2000 was calculated in 1 ton of dry pulp to obtain a concentration of 1 kg of PEG/ton of dry cellulose (typical commercial procedure).

### 2.6. Preparation of Paper Sheets

As a reference sample, a sheet of paper without the addition of inorganic fillers was prepared. The reference paper sample contained only standard commercial components used for the production of paper sheets. Paper with a grammage of 80 g/m^2^ was created by combining three types of cellulose fibers: short-fiber cellulose pulp (eucalyptus derived), long-fiber cellulose pulp (birch-derived), and chemical thermomechanical pulp cellulose (CTMP) with a mass ratio of 70/20/10, respectively. The cellulosic mass was mixed in a plastic container using a mechanical stirrer for 15 min. Subsequently, PAX and a cationic starch solution (3.8%) were added to the mixture with 2 and 5 kg per ton of dry cellulose fibers, respectively. Finally, CaCO_3_ and appropriate crosslinkers (SiO_2_, TiO_2_, h-BN, and h-BN-OH) dispersed in PEG solution were introduced to the system. To do so, 1 kg of crosslinkers per 1 ton of dry cellulose fibers was first dispersed in 100 mL of PEG solution and sonicated to obtain a homogeneous mixture. The prepared mixture was mechanically stirred with cellulosic mass for 15 min. Paper sheets were formed using a Rapid-Köthen Automatic Sheet-Forming Machine (Lodz, Poland), following the guidelines of PN-ISO 5262–2. This method of producing modified paper sheets using the mentioned equipment replicates the conditions found in large-scale production, facilitating easy scalability of the entire process. The prepared paper sheets were denoted as reference for the sample without any crosslinker, and PEG/SiO_2_, PEG/TiO_2_, PEG/h-BN, and PEG/h-BN-OH for samples where additional PEG suspension containing SiO_2_, TiO_2_, h-BN, and h-BN-OH were added, respectively. The reference was prepared by the same procedure but without the addition of PEG/inorganic filler mixture.

### 2.7. Characterization

High-resolution transmission electron microscopy (HR-TEM) (Washington, DC, USA) imaging was performed with the FEI Tecnai F20 microscope at an accelerating voltage of 200 kV. The images were taken directly on sample-drop-cast Cu grids with carbon film. A scanning electron microscope (SEM) (VEGA3, TESCAN) (Brno, Czech Republic) was used to determine the morphology of the prepared sheets. The chemical bonding of the structures in the paper sheets was examined using Raman spectroscopy (InVia Renishaw, Wotton-under-Edge, UK) equipped with an excitation wavelength of 785 nm. It is an ideal method to study the structural properties of the nanomaterials. The phase composition was determined by X-ray diffraction (XRD) patterns by using an Aeris (Malvern Panalytical, Malvern, UK) diffractometer using Cu Kα radiation. The content of CaCO_3_ was determined in accordance with the International Organization of Standardization (ISO 1762:2001). The thermogravimetric analysis was conducted using an SDT Q600 Thermogravimeter (TA Instruments, New Castle, DE, USA) under air flow of 100 mL/min. In each case, the samples were heated from room temperature to 600 °C at a linear heating rate of 10 °C/min. The samples were measured in an alumina crucible with a mass of about 5.0 mg. N2 adsorption/desorption isotherms were acquired at liquid nitrogen temperature (77 K) using a Micromeritics ASAP 2460 (Norcross, GA, USA). The Brunauer–Emmett–Teller (BET) and density functional theory (DFT) methods were adopted to calculate the specific surface area and pore size distribution.

## 3. Results

TEM images of SiO_2_, TiO_2_, h-BN, and h-BN-OH are presented in Figure 1. SiO_2_ and TiO_2_ reveal distinct morphologies. SiO_2_ (Figure 1A) exhibits particles with spherical morphology, showcasing visible porosity within the silica structure. High porosity directly leads to the high surface area of SiO_2_. Similarly, TiO_2_ (Figure 1B) nanoparticles display a spherical or quasi-spherical morphology, relative to SiO_2_, and exhibit observable porosity with evident pores. For h-BN, the TEM image illustrates a flat and two-dimensional (2D) sheet-like structure. It is a layered material with a hexagonal lattice resembling graphene, showcasing a thin and planar 2D nature. Hydroxylated h-BN (h-BN-OH, Figure 1D) also portrays this 2D planar structure, with the presence of hydroxyl (OH) groups altering surface characteristics but not the fundamental structural morphology.

XRD patterns of all crosslinkers (SiO_2_, TiO_2_, h-BN, and h-BN-OH) are shown in Figure 2. Both h-BN and h-BN-OH show reflections corresponding to boron nitride (2θ = ~26.7°, 41.6°, 43.7°, 54.9°, 75.6°; ICDD PDF no. 00-034-0421). For h-BN-OH, a clear shift of peaks toward the higher angles is observed in comparison to h-BN. For example, the signal at ~26.39°, corresponding to the (002) plane of h-BN, is shifted to 26.83°. This is due to the expansion of crystallographic structure by the incorporation of the -OH functional groups into the lattice. SiO_2_ exhibits one broad peak centered at around 23°, which can be assigned to the amorphous structure of silica oxide [17,18]. TiO_2_ is composed of two crystal phases: anatase (ICDD PDF no. 04-014-8515) and rutile (ICDD PDF no. 00-021-1276). There are ~83.9% of anatase and ~16.1% of rutile in the sample [19]. 

N_2_ adsorption/desorption isotherms acquired at liquid nitrogen temperature are presented in Figure 3. The isotherms for TiO_2_, SiO_2_, h-BN, and h-BN-OH are type II isotherms, where there is a wide range of pore sizes [20]. From TEM, it is clear that pores are present. The highest content is in SiO_2_, which can result in a high surface area. The highest surface area was determined for the SiO_2_, which is 275.4 m^2^/g. h-BN-OH exhibits a larger specific surface area compared to h-BN, which is 38.7 m^2^/g and 19.8 m^2^/g, respectively. This is due to the expansion of individual h-BN layers by hydroxyl groups and the creation of a larger surface area that is accessible for N_2_ adsorption. The lowest specific surface area from all crosslinkers was measured for the TiO_2_ (10.8 m^2^/g). A similar dependence can be observed for the total pore volume (Figure 3b). The measured total pore volumes were 0.248, 0.035, 0.011, and 0.008 cm^3^/g for SiO_2_, h-BN-OH, h-BN, and TiO_2_, respectively. Data collected from the N_2_ adsorption/desorption test are collected in Table 1. 

Next, the impact of different crosslinkers on the morphology of the prepared paper samples was evaluated via SEM (Figure 4). The reference sample exhibits the presence of CaCO_3_ between the cellulose fibers with uneven distribution. The PEG/SiO_2_ and PEG/TiO_2_ paper samples (Figure 4B,C) show a more even distribution of CaCO_3_ particles along cellulose fibers compared to the reference sample. In the material with PEG/SiO_2_, the amount of CaCO_3_ between the fibers is much higher compared to the reference sample. Paper samples with h-BN and h-BN-OH exhibit agglomerated CaCO_3_ particles. This may lead to increased permeability of CaCO_3_ through cellulose fibers, which will result in a reduced content of calcium carbonate in the sample. 

Figure 5 presents the Raman spectra of paper samples (reference, PEG/SiO_2_, PEG/TiO_2_, PEG/h-BN, and PEG/h-BN-OH). Cellulose is a linear polymer made up of repeating glucose units linked by β-1,4-glycosidic bonds. It exhibits a high degree of crystallinity, with both crystalline and amorphous regions in its structure. Raman spectroscopy is sensitive to various vibrational modes present in cellulose. The main Raman-active modes for cellulose fibers include C-C and C-O bonds. These vibrations are observed around 1095 cm^−1^, offering valuable information about the molecular bonds within the cellulose structure [21,22]. Next, out-of-plane ring bending (C-C-C and C-O-C), observed in the range of 400 to 600 cm^−1^, provides insights into the spatial arrangement and flexibility of the cellulose rings. Additionally, Raman peaks associated with deformations in the CH_2_ and CH_3_ groups are prominent in the range of 1300 to 1470 cm^−1^ [23] (orange zones). Furthermore, Raman spectroscopy studies revealed the distinctive peaks of CaCO_3_ corresponding to two different forms: calcite (green zones) and vaterite (blue zones). The peak at 1085 cm^−1^ signifies calcite, while the peaks at 1080 and 1090 cm^−1^ represent vaterite, specifically corresponding to the Ag internal mode derived from the v_1_ symmetric stretching mode of the carbonate ion in each material. The v_4_ in-plane bending mode of carbonate is observed at 712 cm^−1^ for calcite and 739–749 cm^−1^ for vaterite. It is noteworthy that the characteristic peak for vaterite was not detected in the case of PEG/h-BN and PEG/hBN-OH. However, the reason of the lack of this peak is not clear.

Figure 6 shows the X-ray diffraction patterns of the reference, PEG/SiO_2_, PEG/TiO_2_, PEG/h-BN, and PEG/h-BN-OH paper samples. In the patterns of all samples, two characteristic phases were identified, i.e., cellulose and calcium carbonate (CaCO_3_). The three observed peaks (broad peaks) at 2θ = 16°, 22°, and 35° correspond to cellulose. However, a series of reflections at 2θ equal to ~23°, 29.4°, 36°, 39.4°, 43.2°, 47.5°, and 48.5° are characteristic for CaCO_3_ (ICDD no. 00-005-2586). The XRD diffractograms also show two peaks (low intensity) at ~30.9° and 31.6°, which can be attributed to other calcium carbonate polymorphs (e.g., aragonite and vaterite) (ICDD no. 01-075-9984, 00-024-0030). The reflections, which correspond to the calcium carbonate, are narrow and intense (compared to the cellulose peaks, the peaks are wider and of lower intensity), which indicates the high crystallinity of the used CaCO_3_. A lack of a significant effect of the fillers on the intensity and location of individual reflections was found, which may be attributed to the small amount of the additives.

To define the thermal behavior of the paper samples, thermogravimetric analysis (TGA) was applied. The TGA results for the reference and the PEG/SiO_2_, PEG/TiO_2_, PEG/h-BN, and PEG/h-BN-OH papers are presented in Figure 7. For all samples, three significant weight decreases are noticeable. First, weight loss is observed at 90 °C and is attributed to the evaporation of the adsorbed water or moisture present in the samples. Furthermore, two stages of cellulose degradation are observed. The first decomposition stage starts at 250 °C. Cellulose consists of glucose molecules linked together by β-1,4-glycosidic bonds [24]. During this stage, glycosidic bonds break, resulting in the release of volatile products and various volatile organic compounds, such as acetic acid and levoglucosan [25,26]. Next, a second decomposition stage starting at 350 °C is observed. During this stage, additional volatile products are formed as cellulose decomposes. Carbon dioxide (CO_2_) and carbon monoxide (CO) are released at this stage as a result of the breakdown of more complex cellulose structures. Starting with the reference, a gradual mass loss with increasing temperature leads to a residue of approximately 26%, indicating the ash content in the reference sample. In the cases of PEG/SiO_2_, PEG/TiO_2_, PEG/h-BN, and PEG/h-BN-OH, the thermal profiles display analogous mass loss curves, resulting in residual masses of about 27.0%, 25.7%, 25.5%, and 24.8%, respectively. 

To determine the ash content in the paper, the guidelines set by the International Organization of Standardization (ISO 1762:2001) were applied. The procedure was through sample ignition at 525 °C. The ash content in the reference and modified papers (PEG/SiO_2_, PEG/TiO_2_, PEG/h-BN, and PEG/h-BN-OH) are presented in Table 2. 

Based on the results presented in Table 2, the addition of SiO_2_ nanoparticles increased the ash content in the paper significantly. This means that this facile strategy boosts CaCO_3_ retention on cellulose fibers by 12.1% in the presence of PEG/SiO_2_ as a nanofiller, which can be related to the abundance of active sites available for CaCO_3_ bonding. The opposite trend can be observed for PEG/TiO_2_, PEG/h-BN, and PEG/h-BN-OH, where ash content is 8.1, 11.1, and 17.5% lower compared to the reference paper, respectively. 

In summary, based on the above results, it can be inferred that the addition of porous SiO_2_ in the form of a PEG suspension is the most promising crosslinker increasing the affinity of CaCO_3_ to cellulose fibers. This conclusion is supported by numerous analyses, such as TGA, and ISO 1762:2001 tests, which showed that the inorganic fillers content after the introduction of PEG/SiO_2_ increased by 12.1%, which proves the potential of this facile strategy in practical application in the paper industry. This may be because SiO_2_ possesses the largest specific surface area among other crosslinkers (TEM and N_2_ adsorption/desorption), increasing affinity to cellulose fibers, resulting in higher inorganic filler content retention. 

## 4. Conclusions

In summary, this study investigated the influence of various crosslinkers (TiO_2_, SiO_2_, h-BN, and h-BN-OH) on the inorganic filler content in paper samples. Raman spectroscopy and XRD confirmed that the chemical structure of the samples remained insignificantly changed upon the addition of these compounds. This study further delved into the molecular dynamics of cellulose through various Raman peaks associated with specific vibrational modes. Additionally, Raman spectroscopy identified characteristic peaks for CaCO_3_, distinguishing between calcite and vaterite forms. Moreover, the introduction of h-BN and h-BN-OH led to the formation of agglomerates and uneven distribution of CaCO_3_ particles on cellulose fibers, resulting in decreased ash content after testing. Among the crosslinkers, SiO_2_ suspended in a PEG solution emerged as a promising candidate due to its excellent affinity to cellulose and high surface area, enhancing inorganic filler content in the paper. These findings contribute to a deeper understanding of how different crosslinkers impact the composition and structure of paper samples and allow the boosting of the content of inorganic compounds in the paper sheet, reducing the contribution of cellulose.

## Figures and Tables

**Figure 1 polymers-16-00110-f001:**
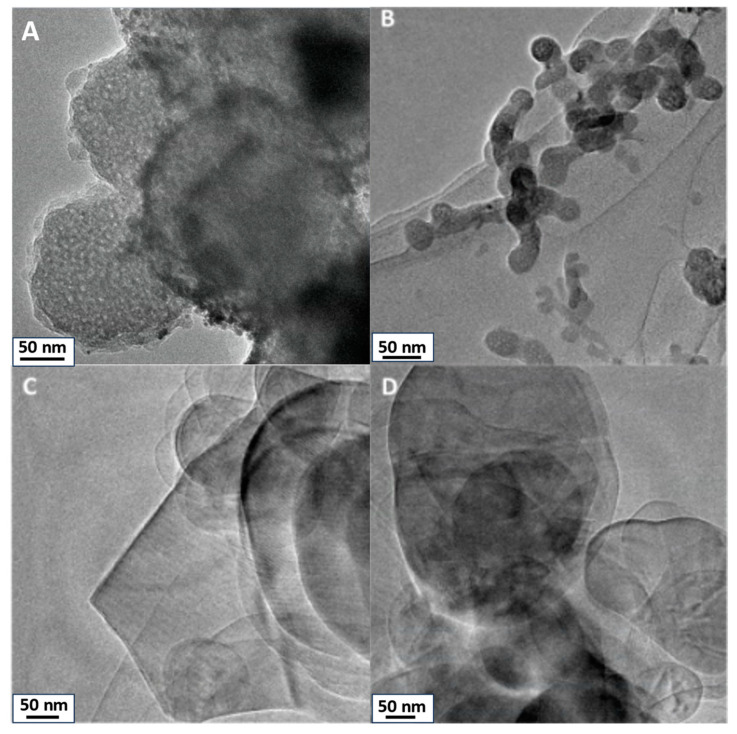
TEM images of crosslinkers: (**A**) SiO_2_, (**B**) TiO_2_, (**C**) h-BN, and (**D**) h-BN-OH.

**Figure 2 polymers-16-00110-f002:**
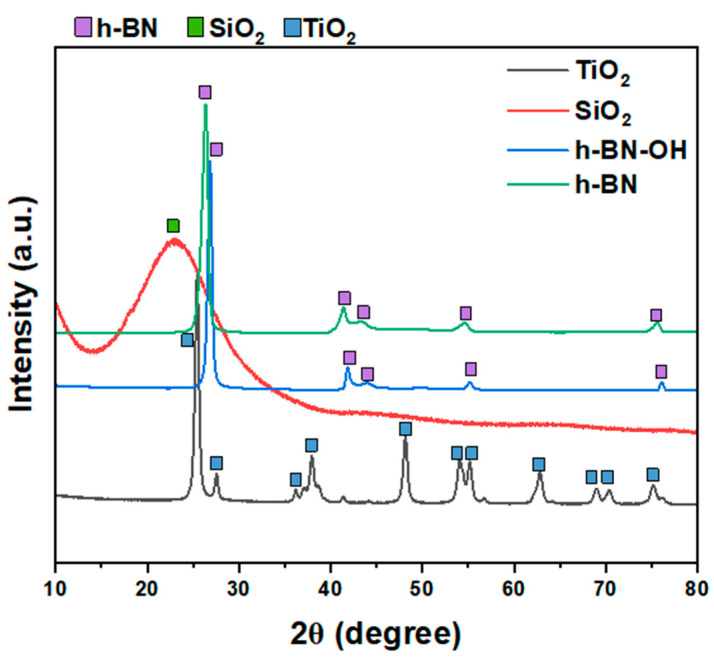
XRD patterns of crosslinkers: TiO_2_, SiO_2_, h-BN-OH, and h-BN.

**Figure 3 polymers-16-00110-f003:**
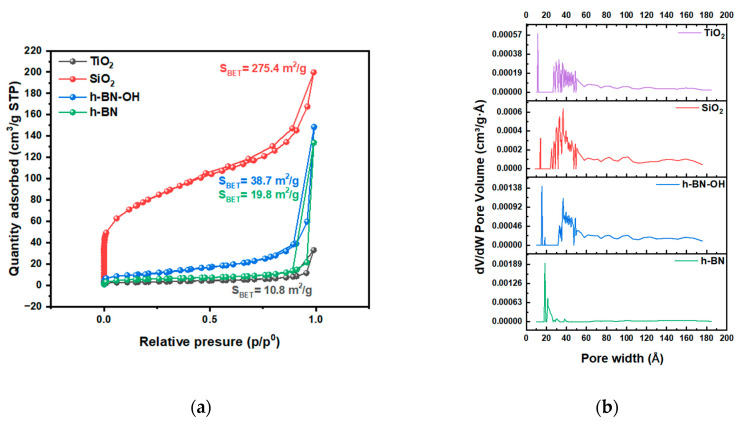
(**a**) N_2_ adsorption/desorption isotherms of crosslinkers, TiO_2_, SiO_2_, h-BN, and h-BN-OH, and (**b**) pore size distribution.

**Figure 4 polymers-16-00110-f004:**
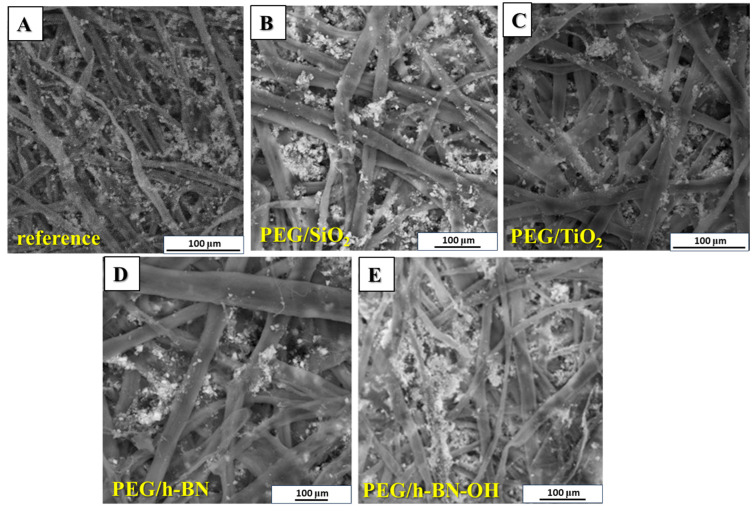
SEM images of (**A**) reference paper and paper containing different crosslinkers: (**B**) PEG/SiO_2,_ (**C**) PEG/TiO_2_, (**D**) PEG/h-BN, and (**E**) PEG/h-BN-OH.

**Figure 5 polymers-16-00110-f005:**
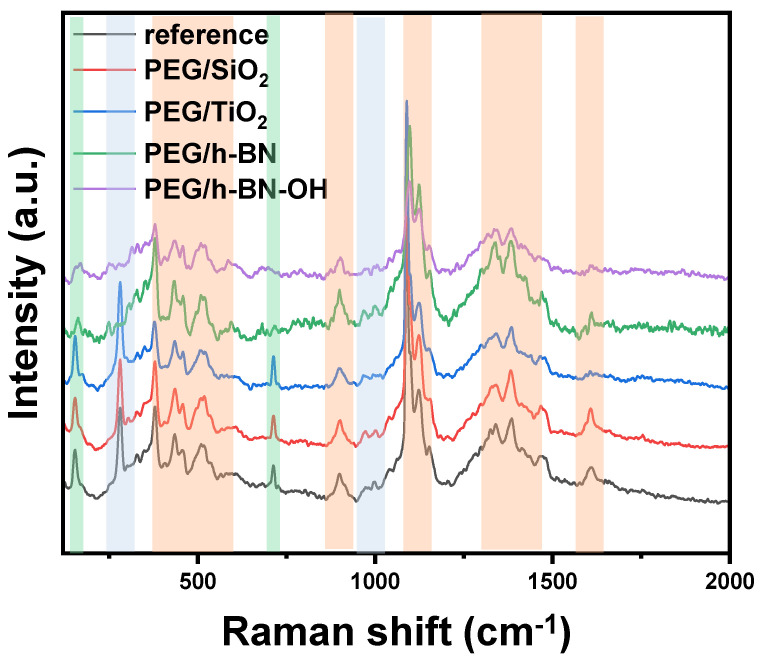
Raman spectra of reference paper and paper with different crosslinkers (PEG/SiO_2_, PEG/TiO_2_, PEG/h-BN, and PEG/h-BN-OH).

**Figure 6 polymers-16-00110-f006:**
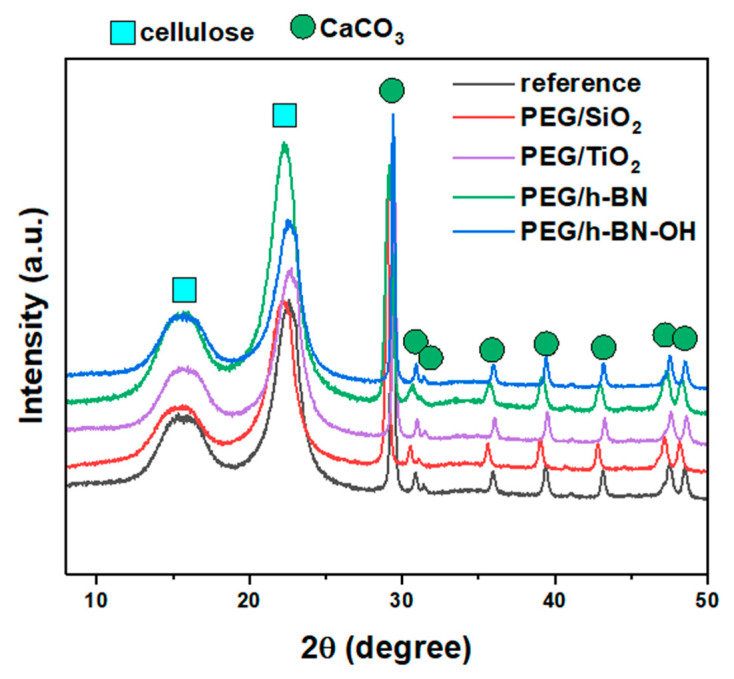
X-ray diffraction patterns of reference paper and paper with different crosslinkers: PEG/SiO_2_, PEG/TiO_2_, PEG/h-BN, and PEG/h-BN-OH.

**Figure 7 polymers-16-00110-f007:**
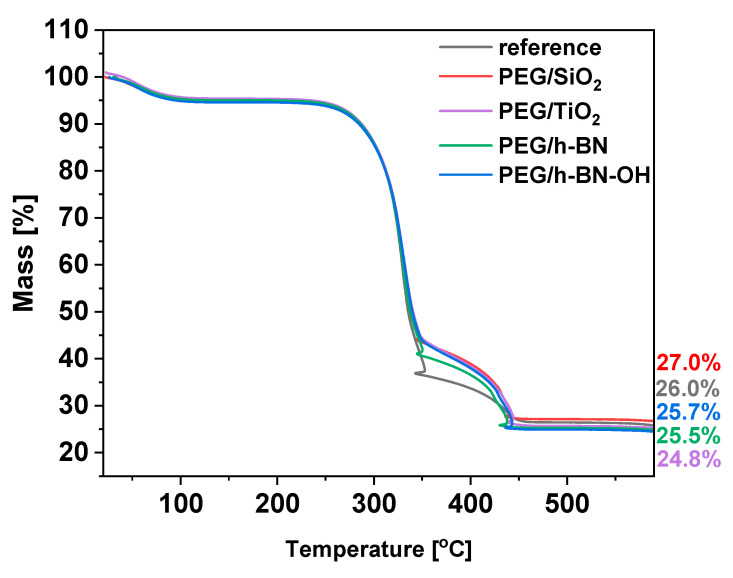
TGA plots of reference paper and paper with crosslinkers: PEG/SiO_2_, PEG TiO_2_, PEG/h-BN, and PEG/h-BN-OH.

**Table 1 polymers-16-00110-t001:** BET surface area, micropore volume, and median pore width of samples.

	BET Surface Area (m^2^/g)	Total Pore Volume (cm^3^/g)	Median Pore Width (nm)
TiO_2_	10.8	0.00772	0.9509
SiO_2_	275.4	0.24762	1.0085
h-BN-OH	38.7	0.03475	1.1414
h-BN	19.8	0.01100	1.2515

**Table 2 polymers-16-00110-t002:** Data obtained from the International Organization of Standardization (ISO) standard (ISO 1762:2001) ash content for all samples.

	Ash Content (%)
Reference	28.1
**PEG/SiO_2_**	**31.5**
PEG/TiO_2_	25.7
PEG/h-BN	25.1
PEG/h-BN-OH	23.2

## Data Availability

The data presented in this study are available from the corresponding author upon reasonable request.

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
