# Peer review of "Facile Strategy for Boosting of Inorganic Fillers Retention in Paper"

_polymers, 2023, doi:10.3390/polym16010110_

Round 1
Reviewer 1 Report
Comments and Suggestions for Authors
The manuscript “Facile strategy for boosting of inorganic fillers retention in paper” requires significant improvements in terms of interpretation of experimental data.
Some comments are presented below.
In the work, the authors use PEG 2000, but there is no explanation why the choice was made in favor of it. Why, for example, was not low molecular weight polyethylene glycol used? Please provide clarification in the text.
More detailed information about the methods should be added in section 2.7. Characterization.
The magnification in picture 1 should be clearer.
I recommend labeling all diffraction peaks (i.e. which planes they correspond to) on XRD patterns. What software was used to describe XRD patterns and compare them with ICDD PDF files for crosslinkers?
Please specify what kind of reference sample you used in the SEM experiments. There is no magnification scale on SEM images.
A very poor analysis of the Raman spectra is made in the manuscript. Please, add information. “With the addition of the crosslinker, the peaks do not lose their intensity, which proves that the cellulose fibers are not damaged.” is an unconvincing argument. For example, the intensity of the Raman peaks between 1200 and 1450 cm-1 have different Imax. Also, no information is provided why the ~700 cm-1 peak disappears in the case of PEG/h-BN and PEG/h-BN-OH. Raman spectra needs to be separated in intensity for better visualization.
The positions of all peaks related to both cellulose and CaCO3 shift along the 2tetta axis when different crosslinkers are added. How can you explain this? How does crystallinity change relative to the reference paper in the case of paper with different crosslinkers? Can you estimate a crystallinity index from XRD data? See for example https://doi.org/10.1186/1754-6834-3-10 or Terinte, Nicoleta & Ibbett, Roger & Schuster, Kurt. (2011). Overview on native cellulose and microcrystalline cellulose I structure studied by X-ray diffraction (WAXD): Comparison between measurement techniques. Lenzinger Berichte. 89.
Line 287: Furthermore, the introduction of each crosslinker did not affect the structural changes of the paper samples. This statement is not true. As a minimum, the parameters of the elemental lattice of calcium change. Obviously, crosslinkers also interact with cellulose (see your XRD data).
The conclusion needs to be rewritten taking into account a more detailed analysis of experimental data, in particular XRD and Raman spectroscopy.
Author Response
thank you very much for your commetn

Reviewer 2 Report
Comments and Suggestions for Authors
In this work, the authors the effects from inorganic crosslinkers including SiO2, TiO2, h-BN, and h-BN-OH, on the interaction between cellulose paper and CaCO3 particles. These inorganic additives were characterized by TEM, XRD, and BET. The paper samples with different inorganic additives were evaluated by SEM, Raman, and TGA. The introduction of SiO2 increased the inorganic fillers content for 12.1 %. This strategy is simple but useful to papermaking process. However, there are a few issues which requires major revisions before acceptance.
1. The authors mentioned several problems in Introduction including “reduced strength and stiffness in the paper”, and “reduced printability” due to the weak interaction between cellulose fibers and CaCO3 particles. This work aims to solve these problems and the ultimate goal should be improve the strength, stiffness and printability of the modified paper, not just improve the filler retention. These properties should be measured and compared.
2. In Figure 4 SEM images, the author claimed “PEG/SiO2 and PEG/TiO2 paper samples (Figures 4B, 4C) show a more even distribution of CaCO3 particles along cellulose fibers compared to the reference sample”. However, it is hard to observe apparent differences of particle distribution in these figures. How did the authors evaluate the distribution?
3. Scale bars are missed in Figure 4 SEM images.
4. The ash content evaluation was guided by ISO 1762:2001. However, this standard was withdrawn. The current standard version is ISO 1762:2019.
5. The ash content for PEG/SiO2 paper sample was increased, but that of PEG/TiO2, PEG/h-BN, and PEG/h-BN-OH modified papers were decreased. The increase “may be because SiO2 possesses the largest specific surface area among other crosslinkers increasing affinity to cellulose fibers, resulting in higher inorganic filler content retention”. What are the potential reasons for the decreases?
Comments on the Quality of English LanguageSeveral sentences are not complete, hard to understand. For example, paper 2 line 59, "creating a more stable."
Author Response
thank you very much for your comment

Round 2
Reviewer 1 Report
Comments and Suggestions for Authors
The authors answered all the reviewer’s questions and comments. Necessary edits have been added to the manuscript.
The manuscript may be published in its current form.
Comments on the Quality of English LanguageMinor editing of English language required